# Isolation of Persister Cells of *Bacillus subtilis* and Determination of Their Susceptibility to Antimicrobial Peptides

**DOI:** 10.3390/ijms221810059

**Published:** 2021-09-17

**Authors:** Shiqi Liu, Stanley Brul, Sebastian A. J. Zaat

**Affiliations:** 1Department of Molecular Biology and Microbial Food Safety, Swammerdam Institute for Life Sciences, University of Amsterdam, 1098 XH Amsterdam, The Netherlands; s.liu2@uva.nl; 2Centre for Infection and Immunity Amsterdam (CINIMA), Academic Medical Centre, Department of Medical Microbiology, University of Amsterdam, 1105 AZ Amsterdam, The Netherlands; s.a.zaat@amsterdamumc.nl

**Keywords:** persisters, isolation, antimicrobial peptide

## Abstract

Persister cells are growth-arrested subpopulations that can survive possible fatal environments and revert to wild types after stress removal. Clinically, persistent pathogens play a key role in antibiotic therapy failure, as well as chronic, recurrent, and antibiotic-resilient infections. In general, molecular and physiological research on persister cells formation and compounds against persister cells are much desired. In this study, we firstly demonstrated that the spore forming Gram-positive model organism *Bacillus subtilis* can be used to generate persister cells during exposure to antimicrobial compounds. Interestingly, instead of exhibiting a unified antibiotic tolerance profile, different number of persister cells and spores were quantified in various stress conditions. qPCR results also indicated that differential stress responses are related to persister formation in various environmental conditions. We propose, for the first time to the best of our knowledge, an effective method to isolate *B. subtilis* persister cells from a population using fluorescence-activated cell sorting (FACS), which makes analyzing persister populations feasible. Finally, we show that alpha-helical cationic antimicrobial peptides SAAP-148 and TC-19, derived from human cathelicidin LL-37 and human thrombocidin-1, respectively, have high efficiency against both *B. subtilis* vegetative cells and persisters, causing membrane permeability and fluidity alteration. In addition, we confirm that in contrast to persister cells, dormant *B. subtilis* spores are not susceptible to the antimicrobial peptides.

## 1. Introduction

Persisters are phenotypic variations of microbial cells that are tolerant to harsh environments such as antibiotic exposure and starvation. Unlike resistant bacteria that rely on mutations, persisters are non-genetic and non-heritable cellular states that can be triggered by certain stresses or generated spontaneously as a bet-hedge strategy [1]. A persistent subpopulation was first found and described in a penicillin-treated population by Hobby et al. in 1942 [2] and subsequently named as “persisters” by Bigger in 1944 [3]. Since then, it was believed that all bacterial species can form persisters [4,5]. The existence of persisters leads to incomplete bacterial eradication both in vitro and in vitro, causing recurrent contamination and infection. Persisters also can promote the formation of antibiotic resistant bacteria through, for example, their survival advantage [6] or increase the spread of antibiotic resistance plasmids [7]. Hence, persister-related studies are of great importance.

Despite their universal occurrence and severe impact, both the origin and characteristics of persisters are still mystifying [8]. This is mainly due to the low frequency of persister in a population and its feature as a transient phenotype variant: persisters can easily switch back to normal vegetative cells after stress removal. The latter problem can be circumvented by the usage of microfluidics and image tracking to analyze persisters whilst exposed to stress conditions and reveal their features at the single-cell level [1,9]. In order to get higher percentage of persisters, stationary phase or biofilm cultures that include more persisters than their exponentially growing counterparts were used in some studies [10]. However, using these two populations to study the formation mechanism or the metabolic characterization of persisters may confound the results with that of slow growing or dying cells [11]. Another attempt is the use of mutants that either generate more toxin, such as overexpressed hipA toxin leading to more *Escherichia coli* persister formation [12], or alter certain metabolic pathway to analyze possible persister formation mechanisms [13]. These methods certainly provided valuable information in terms of the formation or maintenance of persisters. However, the results were often limited to a specific treatment or species [14], which may indicate that persister formation is related to given stressors.

The lack of effective methods for persister enrichment and isolation is another hindrance in persister studies. Long-term and high concentration antibiotic exposure is commonly used as a method to enrich spontaneously generated persisters [15,16,17] or even pre-triggered persisters [18,19]. However, this isolation method itself is already a severe stress condition that can trigger persister formation. In order to avoid this impact, an enzymatic lysis protocol was presented [20]: while the lytic solution quickly (within 20 min) killed normally growing cells with a decreased risk of being a trigger, surviving cells were assumed to be persisters. Nonetheless, this method didn’t provide a solution to separate surviving persister cells from a treated culture. In addition, the surviving cells may also include viable-but-nonculturable (VBNC) cells. Compared with persisters that can quickly resuscitate in normal growing conditions, VBNC cells lose their in vitro culturability. These two phenotypes coexist in both growing cultures and given stress conditions [21,22,23]. Their similar characteristics, such as growth arrested and tolerance, makes them difficult to separate by any isolation method based on metabolic inactivity [24] or antibiotic tolerance [25].

Persisters are great contributors to antibiotic failure. In this case, antimicrobial peptides (AMPs), natural or artificial short peptides that possess broad antimicrobial activity, are intensively researched as promising compounds to kill persisters both in industry and in clinical [8]. The main known mode of action of AMPs against persisters is the non-targeted membrane interruption [26]. For instance, SAAP-148, derived from principal human cathelicidin LL-37, showed high bactericidal activity with enhanced stability against *Staphylococcus aureus* persisters within a biofilm through thinning and permeabilization of the bacterial membrane and a decreased biofilm biomass [27]. In our experiment, the efficacy of two synthetic peptides: SAAP-148 and TC-19 was assessed. TC-19 is also derived from human AMP Thrombocidin-1 [28]. It was found that TC-19 combats *Bacillus subtilis* vegetative cells through gradually dissipating the membrane potential and creating fluid domains in the bacterial membrane [29]. However, the mode of action of TC-19 against persisters is unknown.

In this paper, we use *Bacillus subtilis* as a model strain to generate, isolate, and eradicate persisters with cationic AMPs. *B. subtilis* is a commonly used model species not only for spore forming bacteria, but also for gram-positive non-spore forming bacteria. The aims of this study were: (1) generate and isolate *B. subtilis* persisters; (2) study the efficiency and mode of action of AMPs against *B. subtilis* persisters. We showed that under different stress conditions, different level of persisters and spores were generated, which were subsequently isolated through fluorescent staining followed by cell sorting. Besides, persisters, unlike dormant spores, could be eliminated by AMPs. The peptides used increased membrane permeability and altered membrane fluidity. Overall, our experiments provide the possibility of purifying persister population for further study such as qPCR and microscopy analysis, and in doing so highlight the potential efficacy of AMPs against persisters.

## 2. Results

### 2.1. Exposure to Antimicrobial Compounds Generates B. subtilis Spores and Non-Spore Cells

*Bacillus subtilis* strain PS832 was treated with 100-fold the minimum inhibitory concentration (MIC) of four antimicrobial compounds with different killing mechanisms: 12.5 μg/mL vancomycin (inhibiting the lipid II cycle of cell wall biosynthesis [30]); 6.25 μg/mL enrofloxacin (leading to double-stranded DNA breaks [31]); 100 μg/mL Carbonyl cyanide *m*-chlorophenyl hydrazone (CCCP, a proton-ionophore that causes the dissipation of the proton motive force and then inhibits the activity of bacterial ATPases [32]); and 400 μg/mL tetracycline (reversibly inhibiting protein synthesis [33] and disturbing membrane organization [34]). Time-kill assays showed a biphasic killing curve (Figure 1A), indicating that while most of the vegetative cells were killed by the tested antimicrobial compounds, a subpopulation survived, which matched the definition of persisters [35]. There was no significant difference between a 3-h exposure and a 4-h exposure (*p* > 0.1) in terms of the amount of persister cells generated. For all further experiments we chose a 3-h treatment to generate persisters. Due to the characteristic of *B. subtilis* as a spore-forming bacterium, sporulation could also be stimulated during antimicrobial exposure. Through thermal treatment, we killed all non-spore cells and quantified the number of non-spore cells and spores separately after each antimicrobial exposure. As shown in Figure 1B, different levels of non-spore cells and spores were generated by various treatments (*p* < 0.05).

### 2.2. Double Staining to Indicate Surviving Non-Spore Cells after Antimicrobial Exposure

To mark and isolate non-spore cells for further analysis, 5-(and-6)-carboxyfluorescein diacetate (5(6)-CFDA) and prodium iodide (PI) were used to indicate viability. 5(6)-CFDA is a membrane permeable and esterified fluorogenic substrate. Once 5(6)-CFDA accumulated inside the bacteria and was hydrolyzed by intracellular nonspecific esterases to fluorescent carboxyfluorescein, the charged compound was trapped in the bacterial cytoplasm and showed green fluorescence [36], which has been proven a good functional indicator of a live cell with metabolic intermediates sufficient for activity. PI is a membrane impermeant, red-fluorescent nuclear dye that is commonly used to indicate membrane integrity. Figure 2A is a flow cytometry scheme of double stained samples where Q1 contains 5(6)-CFDA positive and PI negative cells, indicating the surviving population after antimicrobial exposure. *B. subtilis* untreated cells, dead cells and spores were double stained and analyzed (Figure 2B), showing that 5(6)-CFDA staining gave strong green fluorescence in vegetative cells but not in spores and dead cells. In this case, 5(6)-CFDA positive and PI negative cells in antimicrobial-pretreated samples (Figure 2C) were non-spore cells that survived antimicrobial exposure. These cells were coined as likely persisters.

### 2.3. Surviving Non-Spore Cells Identified as Persisters

To confirm that 5(6)-CFDA positive and PI negative non-spore cells were persisters, a subpopulation of cells that are genetically identical to the main population but can survive antimicrobial exposure and regrow after stress removal, MIC measurement and a regrowth experiment after antimicrobial exposure were conducted. MIC values for vancomycin, enrofloxacin, CCCP and tetracycline were re-measured after antimicrobial treatments. No MIC change was observed. This result indicated that the survival of non-spore cells was not due to a resistance-related mutation. A regrowth experiment was carried out by flow cytometry with double stained samples (Figure 3). After inoculating into LB medium, both the fluorescence intensity per cell and the number of 5(6)-CFDA^+^ cells decreased with culture time in untreated sample as well as in antimicrobial treated samples. The result of a negative control, where cells were inoculated into 0.85% NaCl not supporting bacterial growth, clearly showed that the 5(6)-CFDA signal didn’t quench during a 6-h incubation in dark. Thus, the decrease of 5(6)-CFDA with time was caused by bacterial proliferation after stress removal. In summary, the subpopulation in Q1 (5(6)-CFDA positive and PI negative non-spore cells) that survived antibiotic exposure without mutation and that regrew after stress removal, were deemed persisters.

### 2.4. Double Staining and Subsequent Cell Sorting Is an Efficient Method to Isolate Persisters

To isolate persisters for further analysis, fluorescence-activated cell sorting (FACS) was used to sort out the subpopulation in Q1. Microscope images before and after FACS (Figure 4A) showed that 5(6)-CFDA and PI-based FACS analysis and sorting enriched persisters while decreasing the debris present in the samples. To quantify the efficiency of FACS in enriching for persisters, the number and proportion of persisters in one microscope image before and after FACS was analyzed (Figure 4B).

Before cell sorting, most microscope frames didn’t have persister cell. On average, only 1 persister cell was observed in one analyzed image. After cell sorting, on average 78. 6% of the observed cells (10) were persisters (Figure 4B). Note that not all FACS sorted cells had green fluorescence in all images, which was probably due to the fact that the sensitivity of the FACS is much higher than that of the fluorescent microscope [37]. In addition, fluorescence photobleaching during sample preparation and observation may also have led to less 5(6)-CFDA positive cells since microscopy was done after the FACS analysis [38]. In conclusion, CFDA-PI double staining and subsequent cell sorting significantly increased the level of persisters in the samples, making it an efficient method to obtain samples that are highly enriched for persisters.

### 2.5. Stress-Related Gene Expression in Isolated Persisters

To analyze the expression of stress-response related genes of persisters generated under the different stress conditions, relative quantification using qPCR was performed on the isolated persisters. Due to the harsh stress condition of generating persisters, it was important to select a proper reference gene with approximately equal expression levels under different stresses. In this experiment, we selected five commonly used reference genes as candidates to choose a proper reference gene, including *gyrB* (encoding DNA gyrase subunit B [39]), *rrns* (16s ribosomal RNA gene [40]), *gapA* (encoding GAPDH (Glyceraldehyde-3-phosphate dehydrogenase 1) [41]), *rpoA* (encoding DNA-directed RNA polymerase subunit alpha [42]), and *hbsU* (encoding DNA-binding protein HU1 [43]). After comparing the Ct values of each candidate gene, *hbsU* was selected as reference gene (Figure 5). Expression of 5 stress response-related genes was quantified relative to *hbsU* expression (Figure 6), including *liaI* (cell envelope stress [44]), *skfA* (Cannibalism sporulation related stress response [45]), *recA* (SOS response [46]), *ctc* (sig-B regulated general stress response [47]), and *relA* (stringent response [48]).

Upregulated *liaI* indicated that increased cell envelop stress was associated to *B. subtilis* persister formation and/or the maintenance of this persistent physiological state. This result is consistent with earlier studies showing that cell envelope stress was strongly induced by the cell wall targeting antibiotics vancomycin [49]; the membrane-active chemical CCCP [50] and tetracycline [34] also led to increased cell envelope stress response. In response to membrane-damaging agents, LiaH (encoded by *liaI*) contributes to the maintenance of membrane integrity and prevents depolarization in *B. subtilis* [51]. Though the relationship between cell envelope stress and *B. subtilis* persistence is still unclear, our results indicated that increased cell envelope stress is associated with persister formation.

The stringent response, regulated by nucleotide (p)ppGpp, has been regarded as a key role in bacterial persistence in several bacterial species including *B. subtilis* [52]. However, in the present result, the expression of *relA* was decreased in all tested persister samples, especially in tetracycline-triggered persister populations, indicating that *relA*-mediated stringent response is not necessary for persister formation in tested samples. This result is similar to the previous finding that ribosomal antibiotics effectively inhibited *relA*-related stringent response without decreasing bacterial persistent level [53].

The expression of *skfA*, *recA* and *ctc* in tetracycline-triggered persisters was downregulated compared to that in the other three persister groups. Combined with the quantification result that tetracycline triggers lower level of persisters (Figure 1), the result suggested that protein synthesis, which is inhibited in tetracycline-treated cells, is important for persister formation in *B. subtilis*. Interestingly, distinct levels of tetracycline-triggered persisters have been found in different bacterial species. For example, tetracycline treatment significantly increases the persister level in *E. coli* [54], but completely prevents persister formation in *Vibrio cholerae* [55]. These contrasting results show that the persister formation mechanism is highly dependent on the bacterial population used in a given experiment. For vancomycin-, enrofloxacin- and CCCP-triggered persisters, no significant difference in *skfA*, *recA* and *ctc* expression was found (*p* > 0.05), indicating that while sporulation related cannibalism response, SOS response and general stress response might contribute to persister formation, they were not related to the stressors used in the present study.

### 2.6. Cationic AMPs Effectively Kill B. subtilis Vegetative Cells and Persisters but Not Spores

The killing ability of cationic AMPs SAAP-148 and TC-19 was tested toward *B. subtilis* vegetative cells, spores and persisters. For vegetative cells and spores, a drop plate method was used (Figure 7A). Fourteen uM SAAP-148 and 56 uM TC-19 killed vegetative cells within 5 min and longer exposure time didn’t lead to more killing. Besides, SAAP-148 possessed higher efficiency in killing *B. subtilis* vegetative cells compared with TC-19. However, these two AMPs were unable to kill spores even at higher concentration (Figure 7B).

Since persisters and spores co-existed in the antimicrobial-treated samples and AMPs couldn’t kill spores, we used flow cytometry to test the effect of AMPs on persisters instead of the drop plate method. 5(6)-CFDA and PI double staining were used to indicate the viability and permeability of persisters after exposure to the same concentration of SAAP-148 and TC-19, respectively. The flow cytometry results (Figure 7C) clearly showed that 56 uM SAAP-148 and TC-19 killed most persisters generated in different stress conditions within 5 min. However, some of the enrofloxacin-triggered persisters showed higher tolerance to both AMPs, especially to TC-19. With PI staining, increased permeability of persisters was observed after AMP exposure, but cells which were not stained with 5(6)-CFDA nor with PI were observed as well.

### 2.7. Membrane Fluidity Change during Antimicrobial Exposure and Subsequent AMPs Treatments

SAAP-148 and TC-19 both are membrane-targeting peptides [29]. In addition to membrane permeability change, the mode of action of AMPs on *B. subtilis* persisters regarding membrane fluidity alteration was analyzed by laurdan (6-dodecanoyl-2-dimethylaminonaphthalene). Laurdan is a commonly used fluorescent probe to detect membrane phase alteration [29]. The emission spectrum of laurdan changes with surrounding water content due to its sensitivity to solvent relaxation effect [56]. Through calculating the generalized polarization (GP), membrane fluidity change is quantified. After isolation, laurdan-stained persisters were treated with AMPs for 5 min. Laurdan GP microscope images (Figure 8A) and quantified laurdan GP (Figure 8B) clearly showed that both SAAP-148 and TC-19 treatment led to membrane fluidity change, creating fluid patches (red) surrounded by rigidified membrane, while untreated cells showed evenly stained membranes. In addition, even though SAAP-148 presented higher killing efficiency against *B. subtilis* vegetative cells and persisters than TC-19, their abilities of altering membrane fluidity were similar (*p* > 0.05) though patch distribution differed (Figure 8A).

## 3. Discussion

In this paper, we used the model bacterium *Bacillus subtilis* to study the generation, isolation and eradication of persisters. After 3 h exposure to 100-fold MIC of vancomycin, enrofloxacin, CCCP or tetracycline, different level of spores and non-spore cells were generated. These non-spore cells, which were able to survive the tested antimicrobial exposure without resistant-related mutation and regrow after stress removal, were identified as persisters. With 5(6)-CFDA and PI double staining, we were able to stain and isolate persisters from treated cultures through FACS for further studying such as microscopy and qPCR. Then, we found that two cationic AMPs, SAAP-148 and TC-19 showed promising antimicrobial activity against *B. subtilis* persisters and vegetative cells, but not spores. Finally, the mode of action of AMPs against persisters was revealed: both SAAP-148 and TC-19 caused increased membrane permeability and membrane fluidity alteration.

Given that it is a spore-forming bacterium, *B. subtilis* PS832 persisters and spores co-exist upon applying a given set of antimicrobial exposures. In order to quantify persisters and spores after antimicrobial treatment, we used thermal treatment to kill all non-spore cells (which are persisters). The condition used in this experiment, 70 °C for 30 min and ice 15 min, was tested with *B. subtilis* vegetative cells beforehand and isolated persisters afterwards in our experiment which data confirmed that it killed all persisters. This method, although highly efficient, may not suit every putative persister species or strain due to the possible heat resistance of persisters. For example, Masuda et al. [16] showed that compared to vegetative cells, approximately 100-fold more *E. coli* persisters survived 5 min treatment at 57 °C. *Pseudomonas aeruginosa* persisters showed heterogeneous capacity regarding heat resistance and a minority of persisters (ranging from 4–36%) could endure 70 °C treatment for 5 min [57]. Therefore, it is always necessary to ensure that the given stress condition is valid toward persisters since they are much more tolerant than vegetative cells.

The metabolic activity of persisters is a controversial point. Previously, persisters were referred to as dormant or at least with vastly decreased metabolic activity [58], which was also used for persister isolation. For example, Shah et al. [24] isolated cells with undetectable translation levels as persisters. However, only around 30% of the isolated cells could survive antibiotic exposure and regrow on LB agar plates. Actually, persisters were found to be, though growth arrested, metabolically active to some extent [23,59,60]. Here, we used metabolic activity-dependent 5(6)-CFDA to separate persisters from treated samples that also include debris, dead cells, VBNC cells and spores. 5(6)-CFDA positive cells regrew in LB medium after stress removal, indicating that no VBNC cell was mixed with persisters. Our results also provide evidence that persisters are not entirely metabolically inactive, corresponding with a recently proposed “dormancy continuum hypothesis”: persisters and VBNC cells are two closely related subpopulations of a shared dormancy continuum and persisters are more metabolically active than VBNC cells [61]. It should be noted that our isolation method is only suitable for pre-treated cultures where normally growing vegetative cells can be inactivated. The isolation strategy can in principle also be conducted with clinical samples. This is relevant for the isolation of persistent pathogens that cause antibiotic failure and recurrent diseases.

The heterogeneous generation of persisters has been observed in many situations [1,13,62]. Similarly, we also quantified different levels of persisters after various antimicrobial exposures. In the present study, vancomycin triggered formation of more persisters than other tested antimicrobial agents. This phenomenon is worthwhile expanding upon with other bacterial species and glycopeptide antibiotics in further research. Clinically, it will provide more information about whether certain antibiotic classes have a higher potential in triggering persister formation than others. qPCR results also showed relatively different stress responses in each persister population, suggesting that the mechanism of persister formation is related to a given stress condition. The current data opens the way to study the relationship between a stressor used and the persister formation that follows at the molecular level. To do so the various qPCR expression data can be linked in future studies to omics and mutant analyses thus providing molecular mechanistic data on the mode-of- action of each stressor in inducing persister cell formation.

Regardless of growth or metabolic activity, an intact membrane is essential for a persister to stay alive. Therefore, most known anti-persister agents kill persisters through membrane perturbation [8,63]. Similarly, we used isolated persisters to investigate the mode of action of SAAP-148 and TC-19 on persister membrane and found these cationic AMPs rapidly killed persisters, causing increased membrane permeability and altered membrane fluidity. Membrane permeability was detected by a positive PI signal. However, not all persisters killed by these AMPs became PI positive. This may be due to a severely damaged cell membrane caused by excessive AMP exposure leading to a loss of nucleic acids but not of cell structure. The balance of membrane fluidity and rigidity is crucial to maintain membrane activity [64]. Laurdan staining demonstrated that AMPs rapidly disturbed this balance, creating fluid patches surrounded by more rigid membrane. Interestingly not all stressors that we used to generate persisters gave rise to the same membrane fluidity changes in both overall fluidity as well as distribution of high fluidity patches. The reason for the latter may lie in the direct membrane potential effects of the uncoupler CCCP (see ref. [50]) compared to the antibiotics used. A detailed exploration is beyond the scope of the current manuscript.

It has been demonstrated that fluid patches can be induced by AMPs, leading to cytosol leakage and membrane protein dislocation [29]. Membrane rigidification was also found in many AMPs-treated cells [29,65,66]. However, the reason is still unknown. Interestingly, a possible hypothesis is that instead of being a mode of action of AMPs, membrane rigidification is a stress response mechanism of cells against AMP exposure [65].

Taken together, this paper provides an effective persister isolation method for *B. subtilis* that could be extended to other bacterial species and evidences the potential efficacy of AMPs against persisters.

## 4. Materials and Methods

### 4.1. Bacterial Strains and Antimicrobial Compounds Information

*Bacillus subtilis* strain PS832 was used and cultured in 37 °C under continuous rotation at 200 rpm for liquid medium, if not otherwise specified. For each experiment, growth of *B. subtilis* was initiated by plating out a −80 °C glycerol stock onto Lysogeny broth (LB) agar plates. An individual colony was then picked, inoculated into LB liquid medium and cultured overnight. Afterwards, the overnight culture was 1:100 inoculated into fresh LB liquid medium and incubated until an OD_600_ ≈ 0.4 (the early exponential growth phase) was reached. The culture was then diluted with LB liquid medium to an OD_600_ = 0.2 to be used in MIC measurements, time kill assays and antimicrobial exposure experiments as described below. Four antimicrobial compounds were used to generate persisters: vancomycin (Sigma-Aldrich, St Louis, MO, USA), enrofloxacin (Sigma-Aldrich), CCCP (Sigma-Aldrich) and tetracycline (Sigma-Aldrich). AMP TC19 (LRCMCIKWWSGKHPK) [28] and SAAP-148 (LKRVWKRVFKLLKRYWRQLKKPVR) [27] were synthesized by normal 9*H*-fluorenylmethyloxycarbonyl (F-moc) chemistry [67], dissolved in distilled water to 1.2 mM stock solution and stored in −20 °C.

### 4.2. Minimal Inhibitory Concentration (MIC) Measurement

MIC of every antimicrobial compound was determined by measuring the OD_600_ in a microplate reader (Multiskan™ FC, Thermo Scientific, Etten- Leur, the Netherlands). Specifically, 150 μL of culture containing *B. subtilis* at a final optical density OD_600_ = 0.02 and two-fold serial dilutions of antimicrobial compounds were added to each well of the 96 well plate. Control groups included only *B. subtilis* culture without antimicrobial compounds. After 18 h incubation and measurement in plate reader, the minimal concentration that caused non-visible OD_600_ change was determined as MIC. Three biological replicates were performed. MIC measurement was used for (1) determination of the proper working concentration to generate persisters; (2) recheck the absence of resistant *B. subtilis* after antimicrobial exposure.

### 4.3. Time Kill Assay and the Quantification of Spores and Non-Spore Cells

*B. subtilis* was treated with 100-fold MIC of vancomycin, enrofloxacin, CCCP or tetracycline for 3 h (and 4 h only for time-kill assay). Afterwards, cultures were washed with 0.85% NaCl twice to remove antimicrobial compounds. For time-kill assay, total surviving cells in each sample was quantified by 10-fold dilution (ranging from 10^−1^ to 10^−5^) with 0.85% NaCl and then plating 50 μL sample onto LB agar plates. For the quantification of persisters and spores, 200 μL samples were moved into new Eppendorf tubes and treated for 30 min at 70 °C and then on ice for 15 min to kill non-spore cells. Then, 50 μL pretreated samples were plated onto LB plates. After overnight incubation, the number of colonies was quantified and log_10_ CFU/mL was calculated. The number of persisters were the subtracted values of total surviving cells and spores. Noted that the effect of thermal shock on killing non-spore bacteria was checked with *B. subtilis* vegetative cells and afterwards isolated persisters. Three biological replicates were performed.

### 4.4. Flow Cytometry, FACS and Microscopy

Flow cytometry and FACS was performed with the Cell Sorter (FACSAria™ III, BD Biosciences, San Jose, CA, USA). For every sample, unstained cultures containing 1% dimethyl sulfoxide (DMSO, Sigma-Aldrich, St Louis, USA) were used to determine autofluorescent level. A total number of 100,000 events were analyzed if not specified otherwise. Raw data were analyzed with FlowJo (version 10.7.2, FlowJo LLC, Ashland, OR, USA) software. Microscopy was performed with an Eclipse Ti microscope (Nikon, Tokyo, Japan) equipped with phase-contrast and fluorescence components. Microscope images were analyzed by imageJ (National Institutes of Health and the Laboratory for Optical and Computational Instrumentation, University of Wisconsin at Madison, Madison, WI, USA).

### 4.5. Double Staining

In this experiment, we used 5(6)-CFDA (Sigma Aldrich, St Louis, USA) and PI (Thermo Fisher Scientific, Etten- Leur, the Netherlands) to distinguish surviving cells after antimicrobial exposure. 5(6)-CFDA was dissolved in DMSO to make 5 mM stocks and stored in −20 °C. PI stocks were dissolved in DMSO to 20 mM and stored in −20 °C. Three control groups were: untreated early exponential growth phase *B. subtilis* cells; dead cells that were vegetative *B. subtilis* pre-treated with 70% (*v*/*v*) isopropanol for 1 h at room temperature [68]; and *B. subtilis* spores that were cultured, harvested and purified by following the sporulation protocol described by Wen et al. [69]. Antimicrobial-treated samples were pre-treated with 100-fold MIC of vancomycin, enrofloxacin, CCCP or tetracycline. For each sample, after incubation with 50 μM 5(6)-CFDA for 10 min in 37 °C, 30 μM PI was added and incubated in dark for 15 min at room temperature. Afterwards, double stained samples were washed twice to remove remaining dyes for further flow cytometry or microscopy analysis as described in 4.4. 5(6)-CFDA was detected with excitation wavelength at 488 nm and emission wavelength at 519 nm. PI was detected with excitation wavelength at 532 nm and emission wavelength at 617 nm.

### 4.6. Regrowth of Non-Spore Cells

In order to ensure that CFDA^+^-PI^−^ cells are persisters, a regrowth experiment was conducted in addition to MIC remeasurement as described in Section 4.2. After 3 h of exposure to antimicrobials and subsequent staining with 5(6)-CFDA and PI, the dyes were removed from the samples by washing them twice with 0.85% NaCl. Cells not exposed to antimicrobials were used as control. The fluorescence intensity of 5(6)-CFDA in each sample was then measured with flow cytometry, the result of which is shown as *0h*. Subsequently, treated samples and the control were 1:20 inoculated into fresh LB medium while the negative control was 1:20 inoculated into 0.85% NaCl. All cultures were incubated for 6 h. 500 μL culture was sampled to detect 5(6)-CFDA intensity by flow cytometry every hour after 2-h incubation.

### 4.7. qPCR Analysis with Isolated Persisters

To test the expression of stress-related genes in persisters generated in different conditions, real-time PCR (qPCR) was conducted. At least three million persisters were sorted from each antimicrobial treated culture by FACS. After centrifuge at 4000× *g* for 8 min, the pallets were immediately frozen by liquid nitrogen and stored in −80 °C until RNA extraction. RNA isolation was performed with ISOLATE II RNA Micro Kit, Bioline. The concentration and quality of RNA was then measured by NanoDrop ND-2000 (Thermo Scientific). Same amount of RNA was used for cDNA synthesize by iScript™ cDNA Synthesis Kit (BIO-RAD Laboratories, Hercules, CA, USA) based on the instructions. Then, cDNA was diluted with nuclear-free water to a proper concentration based on the concentration of relating RNA for qPCR reaction. 10 uL qPCR reaction was prepared for each qPCR well containing 5 uL SYBR™ Green PCR Master Mix (Thermo Fisher Scientific), 0.4 uL 10 mM forward primer, 0.4 uL 10 mM reverse primer, 1 uL cDNA and 3.2 uL of nuclear-free water. Primers were designed by Primer3 (https://primer3.ut.ee/) or NCBI (https://www.ncbi.nlm.nih.gov/tools/primer-blast/). At least three replicants for each sample and each gene were tested. All qPCR experiments were performed with Applied Biosystems 7300 Real Time PCR System (Thermo Fisher Scientific), and AmpliStar-II 8-Strip 0.2 mL PCR Tubes with Optical Flat Caps (Westburg, Leusden, The Netherlands).

Proper reference genes were firstly selected from commonly used reference genes including *gyrB*, *rrns*, *gapA*, *rpoA* and *hbsU*. To compare the RNA expression level of each candidates, the Ct values were compared directly. The genes that have relatively stable expression levels in every sample were chosen as the reference genes for normalization. For tested genes, the relative quantification was calculated by 2^−ΔΔCT^ method [70]. Tested genes and their primers were listed in Table 1.

### 4.8. Antimicrobial Peptides against B. subtilis Vegetative Cells, Spores and Persister Cells

To analyze the killing effect of tested AMP SAAP-148 and TC-19 on *B. subtilis* vegetative cells and spores, a time-kill assay was conducted based on the previous experiments [71] with some alterations. In brief, early log phase cells and spores with an OD_600_ = 0.2 were prepared and divided into 500 μL aliquots. For each aliquot, SAAP-148 or TC-19 were added to a final concentration from 1.75 μM to 56 μM by two-fold serial dilutions. After 5 or 30 min AMPs exposure at 37 °C, 25 μL samples was moved out and added into 25 μL 0.1% *w*/*v* polyanetholesulfonic acid sodium salt (SPS) to neutralize cationic AMPs. After 5 min incubation at room temperature, 5 μL samples were moved and spotted on LB agar plates. Three biological replicates were performed. The sensitivity of *B. subtilis* persisters to AMPs was analyzed by flow cytometry. Antimicrobial-pretreated samples were first treated with 56 μM SAAP-148 or TC-19 for 5 min, then the same volume of 0.1% *w/v* SPS were added to neutralize cationic AMPs. Afterwards, AMPs-treated samples were double stained with 5(6)-CFDA and PI for subsequent flow cytometry analysis.

### 4.9. Laurdan Staining to Measure the Membrane Fluidity Alteration during Antimicrobial Exposure and Subsequently AMPs Treatment

The fluorescent dye laurdan (6-dodecanoyl-N,N-dimethyl-2-naphthylamine, Sigma Aldrich) is sensitive to membrane phase transition and water penetration [72]. Here, we used laurdan to detect membrane fluidity alteration of persisters after SAAP-148 or TC-19 treatment. The method described by Omardien et al. [29] was followed with minor alterations. Laurdan was dissolved in dimethylformamide (DMF) with a final concentration of 1 mM and stocked in −20 °C. After persister generation and isolation, 10 uM laurdan was added in each sample and incubated at 37 °C for 5 min in dark. Stained cells were then washed with 0.85% NaCl twice to remove remaining dyes. Then, stained cells were treated with 56 uM SAAP-148 or TC-19 for 5 min before microscopy analysis. Laurdan was detected at the excitation wavelength at around 395 nm and the emission wavelength at 470 nm for rigid membrane environment and 508 nm for fluid membrane environment. Laurdan GP of the microscopic images and its quantification were performed using the CalculateGP ImageJ plugin (https://sils.fnwi.uva.nl/bcb/objectj/examples/CalculateGP/MD/gp.html) designed by Norbert Vischer.

## Figures and Tables

**Figure 1 ijms-22-10059-f001:**
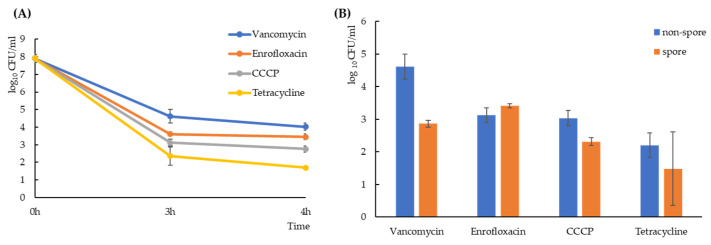
Exposure to various antimicrobials generated different levels (*p* < 0.05) of *Bacillus subtilis* spores and non-spore cells. (**A**) Time-kill curves depicting the total number of surviving cells (including spores and non-spore cells) after 3-h vancomycin (blue), enrofloxacin (orange), CCCP (gray) or tetracycline (yellow) exposure. Quantification was performed by plating treated cultures on LB agar plates and counting colonies. (**B**) The bar-graph shows the number of *B. subtilis* spores (orange) or non-spore cells (blue) after 3 h antimicrobial exposures. The number of spores were assessed after killing all non-spore cells by pre-heating treated cultures in 70 °C for 30 min, placing them on ice for 15 min and subsequent plating. The numbers of non-spore persister cells were then calculated by subtracting the number of spores from the total number of cells cultured after antimicrobial exposure.

**Figure 2 ijms-22-10059-f002:**
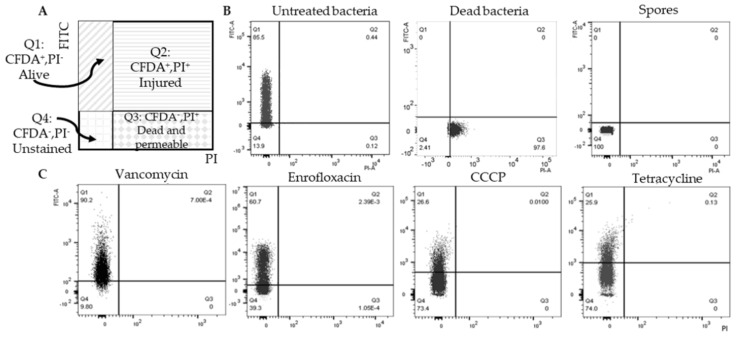
5(6)-CFDA positive and PI negative cells are surviving non-spore cells. (**A**) The flow cytometry scheme of double stained samples; (**B**) Control groups: untreated bacteria (exponential growth phase bacteria), dead bacteria (isopropanol treated) and spores, were double stained with 5(6)-CFDA and PI and then analyzed with flow cytometry. While dead bacteria and spores were not stained by 5(6)-CFDA, Q1 indicated alive non-spore subpopulation in each sample. (**C**) shows flow cytometry patterns of bacteria after antimicrobial exposure and Q1 indicates surviving non-spore cells. 100,000 events in total were analyzed in each sample. This experiment was repeated three times.

**Figure 3 ijms-22-10059-f003:**
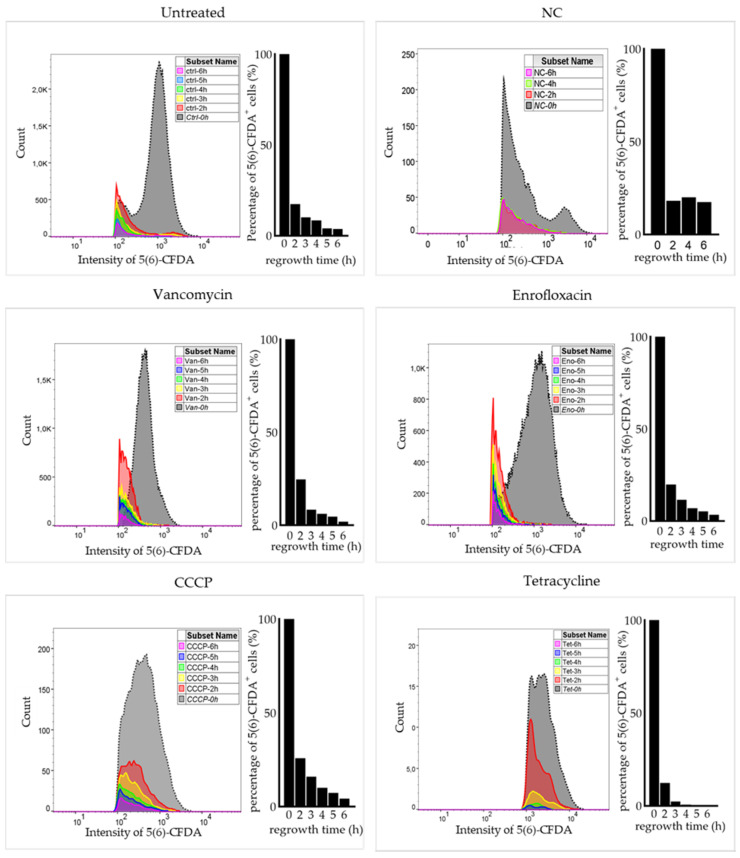
Surviving non-spore cells after antimicrobial treatments are persisters. Surviving non-spore cells after vancomycin, enrofloxacin, CCCP or tetracycline exposure were double stained with 5(6)-CFDA and PI. The fluorescence intensity of 5(6)-CFDA in each sample was then measured with flow cytometry, the result of which is shown as 0 h. Following this, stained cells were 1:20 diluted into LB medium and the fluorescence intensity of 5(6)-CFDA was measured after a 2-, 3-, 4-, 5-, and 6 h incubation. The intensity of 5(6)-CFDA in single cells (**left**) and the percentage of 5(6)-CFDA positive cells compared with the 0 h sample (**right**) are shown for each sample. Results showed that 5(6)-CFDA intensity and the number of 5(6)-CFDA positive non-spore cells gradually decreased during a 6 h incubation in LB medium after stress removal. Compared with the relatively sTable 5(6)-CFDA signal in the negative control (NC) of untreated cells 1:20 inoculated into 0.85% NaCl, the decrease was caused by the regrowth of non-spore cells after stress removal. Note that *0h* was sampled before inoculation and contained more cells than the samples of other time points.

**Figure 4 ijms-22-10059-f004:**
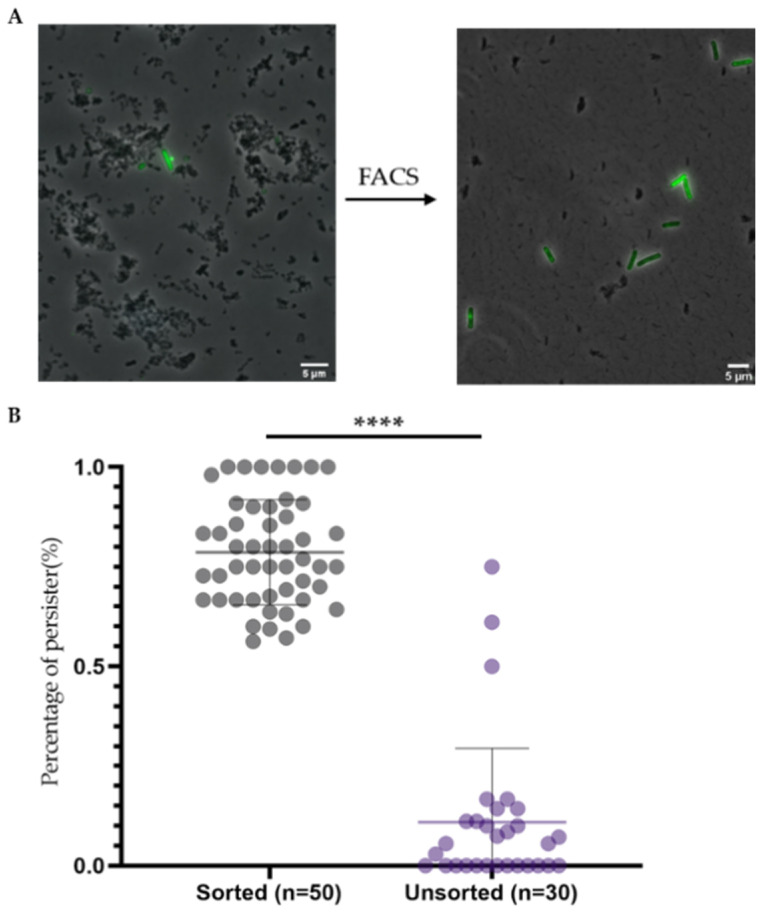
Double staining and subsequent cell sorting significantly enriched persisters. To isolate persisters, antimicrobially treated cultures were firstly double stained with 5(6)-CFDA and PI, then 5(6)-CFDA^+^-PI^−^ cells were sorted with fluorescence activated cell sorting (FACS). (**A**) An example of a merged microscopic image with persisters before and after FACS showed that FACS enriched persisters and decreased the amount of debris at the same time. The images are merged images of phase contrast channel, 5(6)-CFDA channel and PI channel. (**B**) The percentage of persisters as observed in single microscopic images significantly increased after FACS. ****: *p* < 0.0001. On average, in one analyzed image, there was only 1 persister cell before FACS while 10 persisters were observed after sorting.

**Figure 5 ijms-22-10059-f005:**
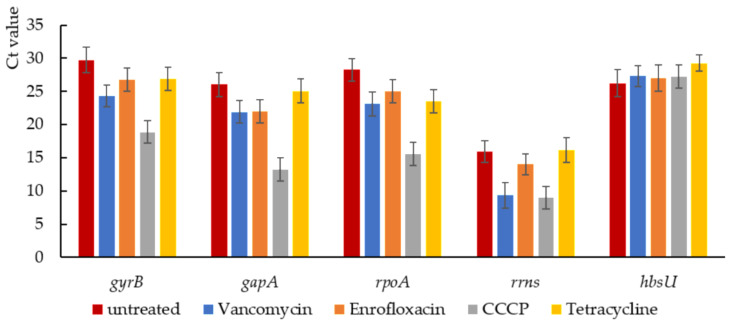
Ct value of five reference gene candidates. In order to choose a relatively stable reference gene for qPCR, expression of five commonly used reference gene candidates were analyzed. In five samples including untreated cells as control and four persister samples isolated from different treatments, the expression of *gyrB*, *rrns*, *gapA*, *rpoA* and *hbsU* were analyzed by qPCR. Result showed that *hbsU* was relatively stable compared to other reference gene candidates.

**Figure 6 ijms-22-10059-f006:**
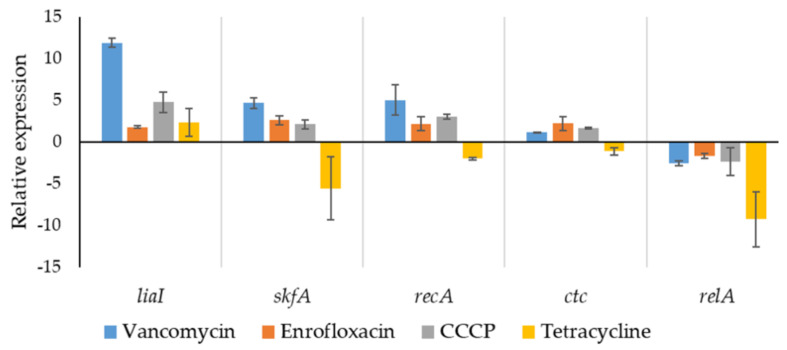
Relative expression level of stress-related genes in isolated persisters. Relative expression of five stress-related genes in isolated persisters generated after treatment with different stressors were determined by qPCR. DNA-binding protein HU1 encoding gene *hbsU* was used as reference gene. *liaI*, regulator of cell envelope stress; *skfA*, regulator of Cannibalism sporulation related stress response; *recA*, regulator of SOS response; *ctc*, regulator of general stress response; and *relA*, regulator of stringent response. *n* = 3.

**Figure 7 ijms-22-10059-f007:**
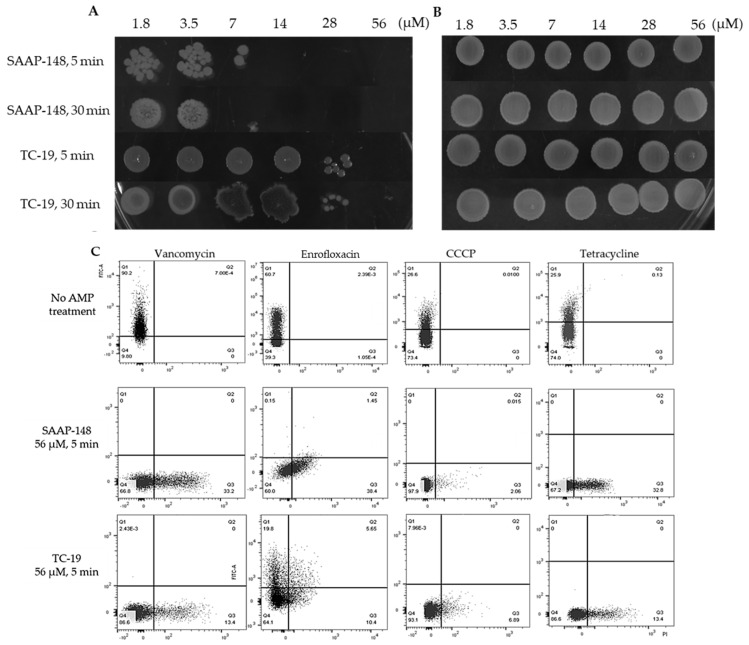
Tested cationic antimicrobial peptides SAAP-148 and TC-19 rapidly killed *Bacillus subtilis* vegetative cells (**A**) and persisters (**C**) with different efficiency, but not dormant spores (**B**). The drop plate method indicated that: (**A**) 14 uM SAAP-148 and 56 uM TC-19 could kill *B. subtilis* vegetative cells within 5 min and elongated exposure time didn’t cause a significant change; (**B**) *B. Subtilis* dormant spores were not sensitive to SAAP-148 and TC-19. To analyze the efficacy of 56 uM SAAP-148 and TC-19 against persisters, 5(6)-CFDA and PI staining was used to indicate the viability and permeability of persisters. Flow cytometry results (**C**) showed that AMPs exposure killed most persisters within 5 min and enrofloxacin-triggered persisters showed higher tolerance to tested AMPs, especially to TC-19. Note that Figure 2C was used in this figure as flow cytometry patterns of bacteria after antimicrobial exposure without AMPs treatment. X and Y axes show the intensity of 5(6)-CFDA and PI, respectively.

**Figure 8 ijms-22-10059-f008:**
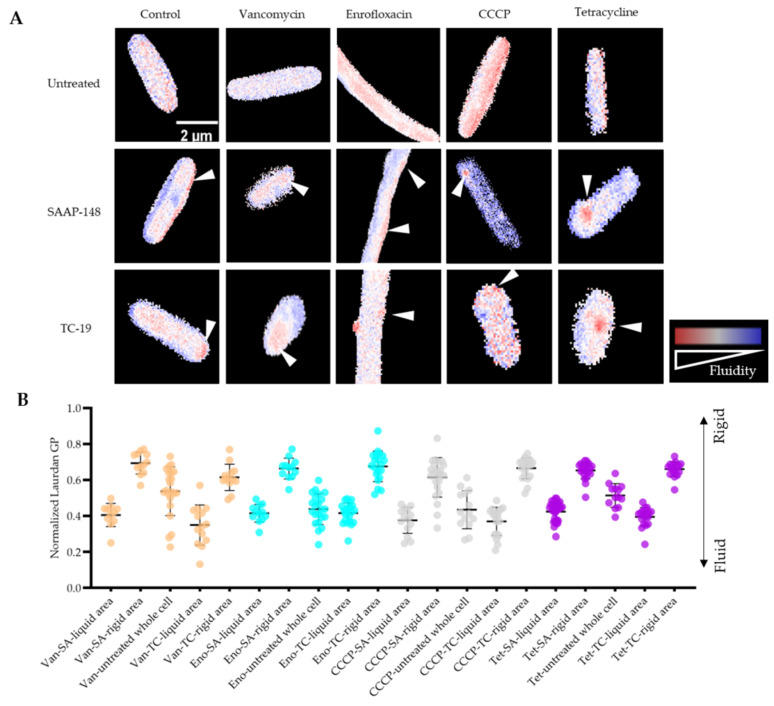
AMPs cause membrane fluidity alterations of persisters. After 5 min SAAP-148/TC-19 exposure, membrane fluidity alteration of persistent and vegetative *B. subtilis* was indicated by laurdan staining. (**A**) Microscope images showed that AMPs exposure created fluidity patch (red, white arrows) on membrane while left parts of membrane became more rigid (blue). (**B**) Average laurdan generalized polarity (GP) of membrane fluidity areas (red) and rigid areas (blue) of persisters resulting from exposure to different stressors was quantified. Compared with the membrane of untreated persisters in each group, fluid area was more fluid (*p* < 0.05), and rigid area was more rigid (*p* < 0.05). No significant difference in the membrane liquid or rigid level between SAAP-148 and TC-19 treated persisters was observed (*p* > 0.05). Van: vancomycin; Eno: enrofloxacin; Tet: tetracycline; SA: SAAP-148; TC: TC-19.

**Table 1 ijms-22-10059-t001:** Primers used for qPCR.

Gene	Forward Primer	Reverse Primer
*gyrB*	GGAGGAAAATTTGACGGAAG	GTTTATAGGTTTGGCGGTGA
*rrns*	AGCATTCAGTTGGGCACTCT	CAGGTCATAAGGGGCATGAT
*rpoA*	GAAGGCGTTGTGGAAGATGT	GCTGCCGTTACAGTTCCTTC
*gapA*	GCTCTTAAAGAAGCGGCTGA	ACCATGCTGCCTTCCATAAC
*hbsU*	TTCCGGCAACTGCGTCTTTA	TGGTAACTTCGAGGTGCGTG
*liaI*	ACAAGAAAACAATAGGCGGA	AACGGAAGTGAGCAGATGA
*skfA*	AGCCGGGAGGTACTTCGATT	AGGATGCGGAAGTGCACAAA
*recA*	GTTCGGCAAAGGTTCCATTA	AGCGCCACAGTTGTTTTACC
*ctc*	TGCAGTCATTACGCTTGAGG	TTCACTCCAATGGCTTCTCC
*relA*	GGCATTGACAACCTCCTTGT	TTCCTTGCGCTTTTGAACTT

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
