# Peer review of "Isolation of Persister Cells of Bacillus subtilis and Determination of Their Susceptibility to Antimicrobial Peptides"

_ijms, 2021, doi:10.3390/ijms221810059_

Round 1

Reviewer 1 Report

The authors of the manuscript described how to generate and isolate so called persister bacterial cells, as well as partially characterized them. The work concerns particularly Bacillus subtilis but can be adapted also for other microorganisms. The study is well conducted, technically sound and presents potentially interesting and novel results in the field of fundamental molecular microbiology. However, before publication the manuscript needs significant revision due to numerous language, spelling, and factual errors. The list of these errors as well as my other comments are presented below:

  1. The title of the paper seems to be awkward and should be modified into a more precise phrase, e.g., “Isolation of persister cells from Bacillus subtilis and determination of their sensitivity to antimicrobial peptides”.
  2. The term “persister cells” concerns particularly microbial cells and not all cells in general. However, this fact is not properly addressed in the Abstract and Introduction. I suggest to modify the appropriate phrases in these sections into the following ones: “Persister cells are growth-arrested subpopulations of microorganisms that can...” and “Persisters are phenotypic variations of microbial cells that are...”.
  3. Line 12: is “persistent pathogens play a key role in antibiotic failure”, should be: “persistent pathogens play a key role in antibiotic therapy failure”
  4. Line 13: is “molecular physiological research into persister formation”, should be: “molecular and physiological research on persister cells formation”
  5. Line 23: is “human cathelicidin LL-37 and human thrombocidin-1 respectively, have”, should be: “human cathelicidin LL-37 and human thrombocidin-1, respectively, have”
  6. Figure 2: please denote the three images in panel B accordingly - as “Untreated bacteria”, “Dead bacteria” and “Spores”.
  7. Line 155: the phrase “a subpopulation that is genetically susceptible” is incorrect.
  8. Line 227: I do not understand what means here “and/or maintenance”.
  9. Line 257: is “Relative expression of stress-related gene in isolated persisters”, should be “Relative expression level of stress-related genes in isolated persisters”.
  10. Figure 7: the lower panels have low quality and are difficult to read.
  11. Line 309: is “Average Laurdan GP of membrane fluidity area”, should be “Average Laurdan generalized polarity (GP) of membrane fluidity areas”.
  12. Line 403-404: Please specify the source of TC19 and SAAP-148 peptides and how these peptides were quantitated before assays.
  13. Chapter 4.2: please specify composition of the buffer/medium as well as the growth phase of the bacteria cells used during MIC doses determination.
  14. Line 408-410: is “Specifically, 150 μl culture, containing B. subtilis cells at a final concentration of OD600=0.02 and a two-fold serial dilution of antimicrobial compounds, was added in 96-well plate”, should be “Specifically, 150 μl of culture containing B. subtilis cells at a final optical density OD600=0.02 and two-fold serial dilutions of antimicrobial compounds were added to each well of the plate”.
  15. Chapter 4.3:  please specify composition of the buffer/medium used during time kill assays.
  16. Line 456-458: is “After 3 hours antimicrobial exposure and subsequently staining with 5(6)-CFDA and PI, dye was removed from samples by washing with 0.85 % NaCl twice”, should be “After 3 hours of exposure to antimicrobials and subsequent staining with 5(6)-CFDA and PI, the dye was removed from the samples by washing them twice with 0.85 % NaCl”.
  17. Lines 458 and 462: the phrases “Control was stained, untreated cells” and “Noted that for 0-hour, stained B. subtilis was sampled before inoculation into LB or 0.85 % NaCl” are incomprehensible.
  18. Line 465: is “stress-related gene”, should be “stress-related genes”.
  19. Lines 471-472: is “Same amount of RNA was used for cDNA synthesize by iScript™ cDNA Synthesis Kit, BIO-RAD based on the introduction. Then, cDNA was diluted with nuclear-free water”, should be “Same amount of RNA was used for cDNA synthesis by iScript™ cDNA Synthesis Kit, BIO-RAD, based on the instruction. Then, cDNA was diluted with nuclease-free water”.
  20. Line 476: is “3.2 ul nuclear-free water”, should be “3.2 ul of nuclease-free water”.
  21. Line 481: is “Proper reference gene was firstly selected from commonly used reference gene”, should be “Proper reference genes were firstly selected from commonly used reference genes”.
  22. Line 483: is “The gene that have relatively stable expression level in every samples was chose as reference gene for normalization. For tested gene”, should be “The genes that have relatively stable expression levels in every sample were chosen as the reference genes for normalization. For tested genes”.
  23. Line 492: is “For each aliquot, SAAP-148 or TC-19 was added”, should be “For each aliquot, SAAP-148 or TC-19 were added”.

Author Response

With respect to its revision, we would like to convey the following, we thank all three reviewers for their comments and criticism. Our rebuttal addresses all items one by one in an itemized list. The revised version contains extensive rewritten sections and has all changes indicated as track-changes.

Response to Reviewer 1 Comments

The authors of the manuscript described how to generate and isolate so called persister bacterial cells, as well as partially characterized them. The work concerns particularly Bacillus subtilis but can be adapted also for other microorganisms. The study is well conducted, technically sound and presents potentially interesting and novel results in the field of fundamental molecular microbiology. However, before publication the manuscript needs significant revision due to numerous languages, spelling, and factual errors. The list of these errors as well as my other comments are presented below:

Point 1:

The title of the paper seems to be awkward and should be modified into a more precise phrase, e.g., “Isolation of persister cells from Bacillus subtilis and determination of their susceptibility to antimicrobial peptides”.

Response 1: We thank the reviewer for his/her views. The title was renamed as “Isolation of persister cells of Bacillus subtilis and determination of their sensitivity to antimicrobial peptides”  

Point 2: 

The term “persister cells” concerns particularly microbial cells and not all cells in general. However, this fact is not properly addressed in the Abstract and Introduction. I suggest to modify the appropriate phrases in these sections into the following ones: “Persister cells are growth-arrested subpopulations of microorganisms that can...” and “Persisters are phenotypic variations of microbial cells that are...”.

Response 2: We amended the text accordingly.

Point 3:

Figure 2: please denote the three images in panel B accordingly - as “Untreated bacteria”, “Dead bacteria” and “Spores”.

Response 3: We amended the figure accordingly.

Point 4:

Figure 7: the lower panels have low quality and are difficult to read.

Response 4: The quality of figure 7 was improved.

Point 5:

Chapter 4.2: please specify composition of the buffer/medium as well as the growth phase of the bacteria cells used during MIC doses determination. Chapter 4.3: please specify composition of the buffer/medium used during time kill assays.

Response 5: Details of medium and the growth phase of bacterial cells were added in line 409-411.

Point 6:

Line 403-404: Please specify the source of TC19 and SAAP-148 peptides and how these peptides were quantitated before assays.

Response 6: AMPs TC-19 and SAAP-148 are synthetic peptides developed and stored by Zaat and coworkers as reported in reference 27 and 28 (Line 77-82 and Line 413-415). They were synthesized by normal 9H-fluorenylmethyloxycarbonyl (F- moc) chemistry, as described in Hiemstra, H.S.; Duinkerken, G.; Benckhuijsen, W.E.; Amons, R.; de Vries, R.R.P.; Roep, B.O.; Drijfhout, J.W. The identification of CD4+ T cell epitopes with dedicated synthetic peptide libraries. Proc. Natl. Acad. Sci. 1997, 94, 10313 LP – 10318. This has been added to Line 413-415.

Point 7:

Lines 458 and 462: the phrases “Control was stained, untreated cells” and “Noted that for 0-hour, stained B. subtilis was sampled before inoculation into LB or 0.85 % NaCl” are incomprehensible.

Response 7: We thank the reviewer for his/her views. These two phrases were rewritten as indicated in Line 468-470.

Point 8:

Line 12: is “persistent pathogens play a key role in antibiotic failure”, should be: “persistent pathogens play a key role in antibiotic therapy failure”

Line 13: is “molecular physiological research into persister formation”, should be: “molecular and physiological research on persister cells formation”

Line 23: is “human cathelicidin LL-37 and human thrombocidin-1 respectively, have”, should be: “human cathelicidin LL-37 and human thrombocidin-1, respectively, have”

Line 227: I do not understand what means here “and/or maintenance”.

Line 257: is “Relative expression of stress-related gene in isolated persisters”, should be “Relative expression level of stress-related genes in isolated persisters”.

Line 309: is “Average Laurdan GP of membrane fluidity area”, should be “Average Laurdan generalized polarity (GP) of membrane fluidity areas”.

Line 408-410: is “Specifically, 150 μl culture, containing B. subtilis cells at a final concentration of OD600=0.02 and a two-fold serial dilution of antimicrobial compounds, was added in 96-well plate”, should be “Specifically, 150 μl of culture containing B. subtilis cells at a final optical density OD600=0.02 and two-fold serial dilutions of antimicrobial compounds were added to each well of the plate”.

Line 456-458: is “After 3 hours antimicrobial exposure and subsequently staining with 5(6)-CFDA and PI, dye was removed from samples by washing with 0.85 % NaCl twice”, should be “After 3 hours of exposure to antimicrobials and subsequent staining with 5(6)-CFDA and PI, the dye was removed from the samples by washing them twice with 0.85 % NaCl”.

Line 465: is “stress-related gene”, should be “stress-related genes”.

Lines 471-472: is “Same amount of RNA was used for cDNA synthesize by iScript™ cDNA Synthesis Kit, BIO-RAD based on the introduction. Then, cDNA was diluted with nuclear-free water”, should be “Same amount of RNA was used for cDNA synthesis by iScript™ cDNA Synthesis Kit, BIO-RAD, based on the instruction. Then, cDNA was diluted with nuclease-free water”.

Line 476: is “3.2 ul nuclear-free water”, should be “3.2 ul of nuclease-free water”.

Line 481: is “Proper reference gene was firstly selected from commonly used reference gene”, should be “Proper reference genes were firstly selected from commonly used reference genes”.

Line 483: is “The gene that have relatively stable expression level in every samples was chose as reference gene for normalization. For tested gene”, should be “The genes that have relatively stable expression levels in every sample were chosen as the reference genes for normalization. For tested genes”.

Line 492: is “For each aliquot, SAAP-148 or TC-19 was added”, should be “For each aliquot, SAAP-148 or TC-19 were added”.

Response 8: Thanks for pointing these out. The above typographical / English style issues have all been addressed.

Reviewer 2 Report

This paper described the development of an effective persister cells isolation method for B. subtilis on the base of persister generation during exposure to antibacterial agents, and isolation of the cells using fluorescence-activated cell sorting. In the second stage of the study, the sensitivity of isolated cells to antimicrobial peptides was investigated. Unlike spores, these cells are found to be sensitive to AMPs that cause membrane permeability and fluidity alteraction. All experiments include corresponding controls, and the data obtained has been statistically processed properly. The work performed is systematic and the conclusions reached by the authors are warranted by the data obtained.

The manuscript is well-written and provided with well-designed illustrative material. I believe that it can be published in its submitted form.

Reviewer 3 Report

In this manuscript. the authors discussed the methods of recovering B. subtilis persister cells, as well as investigated the mode of action of two AMPs on these persister cells. Overall, the manuscript is presented in a great format, regarding the scientific discussion, and the presentation of the results. I would recommend considering this manuscript for IJMS. However, the manuscript still needs some rework and maybe additional data to be accepted. 

Line 107-110: the authors demonstrated the cells cannot be killed under stress matched the definition of persisters. Is there any genomic sequence or plasmid sequence to demonstrate these survived cells are not genetically mutated? Authors should consider whole-genome sequence or at least check plasmid DNA to confirm.

Line 118: the author presented the killing curve of cells under four treatments. The initial inoculum level seems to be at the same number, but it is not clear in the caption, nor in the method description. If the author checked the inoculum level and all conditions have the same initial CFU, it should be explained. Otherwise, if any normalization method was used, authors should discuss that.

Line 151-152: The authors comments that the same number of events were analyzed for each sample. The author should clarify what is "event" referring to, whether the total event or the event in Q1. Also, the authors should provide the total count number to validate the results were based on enough events.

Line 154-168: to further demonstrate the claim and result in this section, can the authors provide the growth curve of Q1 recovered cells to indicate these persisters can still grow in normal LB medium, as well as under the corresponding stress?

Line 197: in order to confirm the cells are alive, authors need to provide fluorescent microscopic images of cells under the PI channel and there should be no PI-positive cell.

Line 217: throughout the manuscript, the authors only discussed 5 genes instead of 6. It is believed "6" is a typo.

Line 242-Line 255, Figure 6: Figure 6 is the gene expression level based on qPCR. Although mRNA is related to protein expression, it is not a direct correlation between gene expression level and protein level. All it is clear is that Tetracycline can down-regulate the expression of 3 genes. This section needs some future clarification. The overall discussion and conclusion of this section is not clear and scientifically sounding.

Line 266: It should be referencing Figure 7A, not Figure 5A.

Line 271-279: None of the results show a more than 50% CFDA- and PI+ population (dead cells), it is not justified to say AMP kill most persisters.

Linne 305, Figure 8A: can the authors explain why the CCCP cell, untreated is very fluidic and became more rigid after AMP treatment? It seems to be different from other conditions.

Line 376-378: the authors claimed that not all AMP killed persisters became PI permeable, and it could be due to the cell lyzing or server membrane damage. This claim is not valid. In the case of cell lyzing, one cannot recover cells for PI staining. There will be no visible intact cells under the microscope. If the membrane is heavily damaged, but still holding the cell structure, the cell will become fully permeable, and it will be PI positive. Neither of the hypothesis the authors raised is valid.

Round 2

Reviewer 3 Report

The authors did a fantastic job addressing the comments for all reviewers. The response and the revised manuscript are well-written. The final manuscript delivered scientific-sounding results and discussion. I believe the manuscript can be accepted in this revised form.